# Cultivating Clinical Clarity through Computer Vision: A Current Perspective on Whole Slide Imaging and Artificial Intelligence

**DOI:** 10.3390/diagnostics12081778

**Published:** 2022-07-22

**Authors:** Ankush U. Patel, Nada Shaker, Sambit Mohanty, Shivani Sharma, Shivam Gangal, Catarina Eloy, Anil V. Parwani

**Affiliations:** 1Mayo Clinic Department of Laboratory Medicine and Pathology, Rochester, MN 55905, USA; 2Department of Pathology, Wexner Medical Center, The Ohio State University, Columbus, OH 43210, USA; nada.shaker@osumc.edu (N.S.); gangal.6@osu.edu (S.G.); anil.parwani@osumc.edu (A.V.P.); 3CORE Diagnostics, Gurugram 122016, India; sambit04@gmail.com (S.M.); shivani.sharma@corediagnostics.in (S.S.); 4Advanced Medical Research Institute, Bareilly 243001, India; 5College of Engineering, Biomedical Engineering, The Ohio State University, Columbus, OH 43210, USA; 6Institute of Molecular Pathology and Immunology of the University of Porto (IPATIMUP), Rua Júlio Amaral de Carvalho, 45, 4200-135 Porto, Portugal; catarinaeloy@hotmail.com; 7Institute for Research and Innovation in Health (I3S Consortium), Rua Alfredo Allen, 208, 4200-135 Porto, Portugal; 8Cooperative Human Tissue Network (CHTN) Midwestern Division, Columbus, OH 43240, USA

**Keywords:** computer vision, digital pathology, whole slide imaging (WSI), artificial intelligence (AI), machine learning, deep learning, diagnostics, laboratory medicine, digital workflow, informatics

## Abstract

Diagnostic devices, methodological approaches, and traditional constructs of clinical pathology practice, cultivated throughout centuries, have transformed radically in the wake of explosive technological growth and other, e.g., environmental, catalysts of change. Ushered into the fray of modern laboratory medicine are digital imaging devices and machine-learning (ML) software fashioned to mitigate challenges, e.g., practitioner shortage while preparing clinicians for emerging interconnectivity of environments and diagnostic information in the era of big data. As computer vision shapes new constructs for the modern world and intertwines with clinical medicine, cultivating clarity of our new terrain through examining the trajectory and current scope of computational pathology and its pertinence to clinical practice is vital. Through review of numerous studies, we find developmental efforts for ML migrating from research to standardized clinical frameworks while overcoming obstacles that have formerly curtailed adoption of these tools, e.g., generalizability, data availability, and user-friendly accessibility. Groundbreaking validatory efforts have facilitated the clinical deployment of ML tools demonstrating the capacity to effectively aid in distinguishing tumor subtype and grade, classify early vs. advanced cancer stages, and assist in quality control and primary diagnosis applications. Case studies have demonstrated the benefits of streamlined, digitized workflows for practitioners alleviated by decreased burdens.

## 1. Introduction

Nearly 2000 years have passed since Emperor Marcus Aurelius sought reinforcement for a society decimated by the first wave of the deadliest pandemic to impact ancient Rome. The same factors lauded as strengths for the seemingly impenetrable empire, e.g., expansive trade networks and large, crowded populations, were those which ultimately led to its demise. These precarious elements had long lingered as a silent plague within a territorial superpower fully primed to combat the fiercest of invaders, yet one which succumbed to those overlooked behind its volcanic rock fortifications. A hidden tinderbox of similar proportion was ignited to plume within many pathology departments upon inception of the 2019 coronavirus (COVID-19) pandemic [1]. New safety and practice restrictions following the wake of the pathogen’s propagation increased the demand for digital pathology (DP) solutions and remote services. Issues that had lingered throughout many departments were fervently exacerbated, e.g., specialist deficits and demands of shorter turnaround times (TAT) amidst increasing caseloads and complexity of pathology reports for aging patient demographics harboring higher disease incidence. New solutions were necessitated upon the exhumation of long withstanding problems [2,3]. Diagnostic surgical pathology remains the ‘gold standard’ for cancer diagnosis despite substantial inter-observer variability from human error, e.g., bias and fatigue, leading to misdiagnosis of challenging histological patterns and missed identification of small quantities of cancer within biopsy material. Digital (whole slide) imaging, now synonymous with DP, has achieved significant milestones within the last 20 years, with whole slide image (WSI) scanning devices evolving in tandem with challenges pervasive throughout the modern pathology landscape. Batch-scanning and continuous or random-access processing capabilities enabling the concurrent uploading of glass slides during the image capture and digitization processes of others have improved laboratory efficiency [4,5]. Many WSI devices can now handle an array of mediums cast on slides of varying dimensions, with single slide load capacity of some devices reaching up to 1000 [2]. WSI scanning cameras and image sensors deliver superior sensitivity, resolution, field-of-view (FOV), and frame rates for optimal capture and digitization of glass slide specimens [2]. Newer scientific CMOS (sCMOS) sensors are featured in many current WSI scanning devices, often as adjunctive to multiple CCD and CMOS sensors for optimization of image quality. 

The Ohio State University (Columbus, OH, USA) was among the first academic institutions to invest in DP devices initially purposed for research and archival, i.e., retrospective scanning of oncology cases [6]. Complete transition to a fully integrated digitized workflow for primary diagnosis followed one year after initial steps toward DP adoption in 2016 (Figure 1).

Beneficial returns from the preemptive digital transformation were evidenced throughout the first wave of the coronavirus pandemic in 2020, during which the department was well positioned to continue educational and research activities with minimal disruption [6]. Clinical services persisted with relative fluidity with digital workflow emerging as a pillar of stability during an otherwise catastrophic downtime event for many. Temporary remote sign-out authority issued by the Centers for Medicare and Medicaid Services (CMS) emphasized a growing acknowledgment of the utility that digital practice may afford during such times. The substantial percentage of pathologists (71.4%) who were already trained and approved for on-site WSI for primary diagnosis at the department increased to 90.6% during the pandemic (reflecting a conglomerate percentage of pathologists using WSI exclusively for primary diagnosis and those using WSI in conjunction with glass slides). Diagnostic quality assurance (QA) evaluation noted little discrepancy pertaining to the percentage of major and minor diagnostic errors accrued prior to and following the viral catalyzation of digital workflow. Intraoperative consultation services also remained considerably unaffected from digital deployment. Real-time rerouting of slides to available pathologists in different locations increased staffing flexibility. Loosening of work-from-home restrictions including sign-out fostered greater pathologist latitude. Reduction in in-person interactions and the number of individuals handling case materials served to reduce viral transmission while also reducing glass slide contamination potential. An aging population of pathologists at the department, reflective of US specialist demographics, reported greater satisfaction from improved office ergonomics following DP implementation, e.g., forward screen-viewing fostered a more natural reading position in comparison to microscopy techniques requiring bending movements [3]. Lastly, WSI viewing software equipped with tools for WSI annotation, precision measurements, and side-to-side WSI viewing programs with virtual magnification and annotation tools enabled pathologists to effectively collaborate via image sharing and real-time slide examination mimicking laboratory conditions despite working from remote locations. WSI viewing software also facilitated comparison of H&E images to corresponding immunohistochemistry (IHC) or special stained slides, further aiding the ease and efficiency of intradepartmental consultations. 

Diagnostic merits of WSI are evidenced in scores of investigations reporting significantly high concordance rates with conventional microscopy throughout numerous disciplines and increasingly for arenas formerly posing hurdles curtailing digital adoption, e.g., cytopathology [7,8,9,10]. Obstacles in modeling business viability for laboratory digitization are surmounted as advanced technology enables a similar roadmap to ubiquitous DP diagnostics already traversed by radiology [10]. The interconnectivity of pathologists, staff, and resources observed following WSI implementation at The Ohio State University reflect a primary endpoint of laboratory digitization. Augmentation of DP tools with artificial intelligence (AI)-based algorithms reflect another. As the university’s primary diagnostic novelty recedes amongst a growing global normalcy of automated workflows, increased efforts for diagnostic quality, and the creation of integrated ecosystems supportive of computational pathology [2,11,12,13], further capitalization from digital integration is now within reach for digitized departments primed to actuate the clinical potential of predictive diagnostic AI-technology for WSI. 

## 2. Development of Computer Vision for Pathology

Computer-aided image processing and pattern recognition, e.g., classification, of histological and cytological structures for pathology has developed from the early 1970s [14,15].

Primordial AI tools for pathology classification tasks typically find genesis at the same vantage point from which modern machine learning (ML) tools began their evolution. Pixel-based analysis, e.g., computer-recognition of a unique series of numerical values that form a shape of interest, is used for classification, e.g., segmentation, tasks that are now among the most essential applications included within integrative workflow image analysis (IA) tools. Traditional morphometric feature evaluation entailed calculation of object size via computational counting of pixels occupied by an object followed by calibration for magnification [16]. Description of object shape resulted from computer determination of a specific shape from a rigid set of preprogrammed rules. Traditional programming directives utilized shape descriptors, e.g., elongation factor and nuclear roundness factor, to identify structures such as peripheral blood erythrocytes. Substantial focus has been directed toward development of computational IA for genitourinary (GU) pathology. Prototypal quantitative light microscopy applications for urological oncology were initially applied to histological sections for rudimentary tumor recurrence and grading predictions [16].

As evidenced from early explorations in computer vision for pathology, traditional programming methods were inherently prone to rapid devolution when image shapes did not adhere to specific pre-programmed rules/definitions, thereby confounding the narrow window of computational interpretability allotted through the ridged training modus. For example, nuclear roundness factor (NRF), defined as the ratio of an area to a perimeter, was observed to decrease when an object shape, e.g., ellipse, deviated from congruence with a perfect circle. The restrictive nature of the programmed code for NRF had predisposed it to conflating “roundness” with “circularity”.

ML techniques have widened the window of interpretability through algorithmic modeling via the use of images rather than preprogrammed rules as input data for algorithm training, allowing computers to correctly visualize shapes regardless of their size, symmetry, or rotation. ML for computational pathology has enabled the interpretive ability of algorithmic tools to extend beyond the limited output yielded from cast-iron programming codes to a system that is able to deduce patterns with increasing accuracy through training. Most current ML approaches utilize methods such as “Random Forest classification”, an algorithmic approach developed in 2001 (the same year as the Leica Aperio T1 gained distinction as the first WSI device released for commercial use) by which a series of decision trees are employed to make an aggregated prediction (Figure 2) [17].

Machine learning allows computers to recognize patterns and make predictive decisions without explicit, step-by-step programming. Trial, error, and extensive “practice” are the core elements of model building for ML, the essence of which follows an iterative approach akin to flashcard memorization. Algorithms are fashioned from a point of zero training data through learning from an output set. A preselected group of image/shape descriptors are chosen by a computer, initially at absolute random, to describe input data fed into the system by a developer. An incorrect label ascribed to an input image by a computer will be amended to display the correct image description from which a machine may demonstrate its learning capacity via correct attribution of the label to a future input image. The ML system takes account of every image pixel and its surrounding pixels with each estimation to ultimately build its own set of rules/algorithms, progressively fashioning an adroit apparatus for predictive accuracy and precision as the cycle continues. Predictive classification models may be tuned and optimized via additional data input providing more opportunities for improvement through trial-and-error for increased accuracy of pattern recognition within new images.

Deep learning (DL) has further expounded upon the cognitive model of ML algorithms, achieving remarkable mimicry to the neural network of the human brain. Artificial neural networks (ANNs) consisting of weighted, interconnected nodes comprise the scaffolding of DL modeling for pathology (Figure 3). 

Powerful neural networks contain up to millions of nodes arranged in layers including input layers, hidden layers, and output layers (Figure 4).

Outputs from one layer of a neural network act as inputs which feed into the nodes of another layer. Convolutional neural networks (CNNs) are a complex derivative from the ANN model fashioned for outcome prediction from WSI data inputs without the assistance of a predefined output set. CNNs for WSI analysis have demonstrated substantial capacity to effectively aid in primary diagnostic and quality control (QC) applications. Other DL models such as the recurrent neural network (RNN) may be used to enhance CNN analysis through provision of spatial and contextual modeling enabled from a bi-directional framework equipped to process high-resolution gigapixel WSIs without image-patch modeling techniques suggested to compromise overall tumor size and sub-structures present within a WSI (Figure 5).

CNN models may use WSIs ascribed individual diagnostic target-labels per associated pattern, e.g., Gleason grade, or up to millions of unlabeled WSI image-patches for autodidactic training during which the AI-model will learn to identify and extract important features without developer assistance (Figure 6).

“End-to-end” methods for training DL models for WSI have greatly mitigated and outperformed highly supervised, effort-intensive methods of algorithmic training dependent upon manually annotated, pixel-based feature extraction techniques (Figure 7).

Algorithm development is typically divided into a series of steps beginning with procurement of clinically annotated samples followed by WSI annotation. An algorithm is developed via a training set and tested via an independent validation set (Figure 8).

A clinical cause pertaining to a relevant population of interest formulates the origin and endpoint for algorithm development driven by a computational pathology team. Pathologists act to direct the genesis and culmination of clinical algorithms while data scientists, e.g., statisticians and bio-informaticians, assist in algorithm design and training. Engineers maintain hardware and software for the operating environment. Pathologists invoke downstream development though providing context through clinically relevant questions that spearhead algorithmic solutions. They are essential for the verification and validation processes for application and monitoring of the algorithm prior to and following clinical deployment, such that feedback is relayed to developers for optimization (Figure 9).

## 3. Realizing the Clinical Potential of AI

The potential for AI to catalyze clinical transformation has been exemplified through recent research, academic, and translational investigations in algorithm development for predictive diagnostic and prognostic analysis made directly from H&E-stained WSIs [18]. Such studies, indicative of the potential for AI to enhance pathologist understanding of disease and improve patient quality of care, encourage further investigations where algorithms may be deployed and evaluated within standardized settings. AI for primary diagnostic and quality control applications may be optimized through clinical trials. Algorithmic development for prostate cancer needle biopsies [19,20,21,22,23,24,25,26,27,28], radical prostatectomies [29,30], and tissue microarrays [21,31,32,33], has held the brunt of focus for the transition of such tools into utilization within clinical forums thus far. Though such investigations have shown promise for AI-assisted grading for prostate cancer and pathologist-review, many have been susceptible to biases and limitations during both development and validation processes, many of which affect the clinical translatability of algorithms developed within non-clinical, e.g., research, settings. The most prominent hurdles affecting clinical implementation of ML and DL tools stem from data availability, generalizability, and transparency (“black box”) concerns (Figure 10).

### 3.1. Overcoming Inter-Observer Variability for Challenging Diagnosis with AI

Challenging histology and morphology is often met with enduringly high rates of inter- and intra-observer variability and increased time-to-diagnosis from pathologists using light microscopy [34]. Discordance is further emphasized within the focuses of genitourinary and renal pathology, where interpretation of complex grading systems, e.g., Fuhrman and Gleason, and prognostic patterns, e.g., cribriform and glomerulosclerotic, is concerningly incongruent even amongst specialists [35,36,37,38,39,40,41,42,43,44,45,46,47,48]. Inter-pathologist grading assessments for prostate cancer grading have elicited concerning results, with kappa values reported as low as 0.3 [38,49]. The last 7 years of ML development for prostatic adenocarcinoma has yielded results demonstrating potential for greater diagnostic objectivity.

2016 marked the first account of DL network development for the detection of prostate adenocarcinoma in core needle biopsy (CNB) tissue. Slide patches extracted from H&E-stained prostate biopsy tissue slides from 254 patients were separated into training, testing, and validation sets. Mean ROC for the median analysis was 0.98–0.99 for the 90th percentile analysis [19].

An advanced CNN derivative was trained for prostate cancer grading using 0.6 mm diameter cores from primary prostate carcinomas in TMAs from 641 patients and tested using TMA cores from 245 prostatectomy cases from another cohort graded independently by two pathologists. Agreement between the DL model and each pathologist was 0.75 and 0.71, respectively, per Cohen’s quadratic kappa statistic, with an inter-pathologist agreement of 0.71. Furthermore, the model demonstrated significantly greater accuracy in distinguishing low-risk from intermediate-risk (*p* = 0.098) cancer than either pathologist (*p* = 0.79 and *p* = 0.29, respectively) [32].

A total of 752 tissue biopsies from multiple sites were used to train a DL system for Gleason grade (GG) identification. Model agreement with pathologists was 72% (68–75%) for specialists and 58% (54–61%) for general pathologists. The model was less likely to over-grade WHO grade group 1 than grade group 2 and more likely to undergrade higher grades in comparison to general pathologists. ROC curves distinguished model-based grade groups 1 and 2 from grade groups 3 through 5 (AUC = 0.97) [50].

Another study in which a CNN was trained for GG classification using 5759 biopsies from 1243 patients yielded a kappa value of 0.85 when compared to three genitourinary pathologists, superior to the kappa of 0.82 obtained from a pathologist panel [51].

Corroborating the potential for AI to improve pathologist grading of prostate biopsies, Bulten et al. recruited fourteen genitourinary specialists to evaluate 160 biopsies with and without assistance of AI algorithms. Using AI, the panel of pathologists demonstrated significantly greater agreeability, yielding kappa values of 0.87 vs. 0.799 when graded independently [52].

ML tools have recently seen development for the automated detection of cribriform pattern in prostate WSIs [41,46,47]. The first instance of ML applications applied to investigate the prognostic utility of invasive cribriform adenocarcinoma (ICC) within specific Gleason grade groups provided insight on the strong prognostic role of ICC morphology fraction of tumor area (cribriform area index(CAI)) in patients with Gleason grade 2 cancer due to the morphology conferring a higher concordance index for biochemical recurrence than patients without evidence of ICC. A CAI increase by a factor of two was determined to be prognostic in patients with ICC morphology after controlling for Gleason grade, surgical margin positivity, preoperative prostate-specific antigen level, pathological T stage, and age (hazard ratio: 1.19) [47].

AI-approaches have demonstrated the capacity identify subtle morphological differences, e.g., sarcomatoid vs. spindle cell pattern, in clinical groups of patients with clear cell renal cell carcinoma (ccRCC) [34]. ML-models have demonstrated the ability to classify early vs. advanced stages of ccRCC, with recent algorithms using gene expression profiling to classify ccRCC stages. One study analyzed gene expression of 523 samples to identify genes differentially expressed in early and late stages of ccRCC, achieving a maximum accuracy of 72.64% and 0.81 ROC using 64 genes on validation dataset [53].

Fenstermaker et al. [54] developed a CNN model to detect, grade (Fuhrman 1–4), and distinguish RCC subtypes (clear cell, chromophobe, papillary). The model was trained on 3000 normal and 12,168 RCC H&E-stained tissue samples of RCC from 42 patients (acquired from the Cancer Genome Atlas). The model classified normal parenchyma vs. RCC tissue with 99.1% accuracy, demonstrating an additional 97.5% accuracy in distinguishing RCC subtypes. Model accuracy in predicting Fuhrman grade was 98.4%.

Two studies using ML models developed from features extracted from single and multi-omics data for classification of early and late stages of papillary RCC emphasized the utility of model-training from multiple data sources. Gene expression and DNA methylation data were used in the later (2020) study, demonstrating slightly better predictive performance than the former (2018) study (MCC 0.77, PR-AUC 0.79, accuracy 90.4) [55,56,57]. A total of 104 genes from Cancer Genome Project expression profiles of 161 patients were used as data in both studies.

Misdiagnoses may lead to delays in appropriate treatment regimens for patients presenting with challenging morphology that is often misidentified. The subtle morphologic characteristics which differentiate the TFE2 Xp11.2 translocation variant of RCC (TFE3-RCC) from other RCC subtypes often leads to the misdiagnosis of this aggressively progressive form of RCC and was the basis for a recent ML development for its identification. An automated ML pipeline was developed to extract TFE3-RCC features and used to differentiate subtle morphological differences between TFE3-RCC and ccRCC with high accuracy. AUCs ranged from 0.84 to 0.89 when evaluating classification models against an external validation set [58].

### 3.2. Exploring AI Development for Nephropathology Applications 

Glomerulosclerosis and IFTA are histologic indicators of irreversible kidney injury, with cortical fibrosis holding distinction as the single greatest morphologic predictor of chronic kidney disease, regardless of disease etiology [59]. Quantification of glomeruli and glomerulosclerosis on kidney biopsy are among the constituents of a standard renal pathology report, yet the prevailing methods for glomerular assessment remain manual, labor intensive, and non-standardized [60]. Although manual evaluation of glomerulosclerotic percentage has consistently demonstrated high inter-observer concordance, traditional visual quantitation of renal cortical involvement incurred by IFTA results in higher variability among pathologists due to the innately complex histology and diverse morphology of the region [59].

The first CNN fashioned for multiclass segmentation of digitized periodic acid-Schiff (PAS) stained nephrectomy samples and transplant biopsies indicated the necessity for more studies interrogating quantitative diagnostic tools for routine kidney histopathology [61]. Significant correlation between pathologist-scored histology vs. the CNN was noted for glomerular counting in whole transplant biopsies (0.94 mean intraclass correlation coefficient). The CNN yielded the best segmentation results for glomeruli in both internal and external validation sets (Dice coefficient of 0.95 and 0.94, respectively), with the model detecting 92.7% of all glomeruli in nephrectomy samples.

The nephropathology landscape has provided fertile grounds for the development of ML tools fashioned to parse and delineate various complex morphological structures, as demonstrated in a slew of recent investigations suggesting the clinical merit of AI within the medical kidney arena [62]. CNN-directed segmentation of morphologically complex image structures, e.g., interstitial fibrosis and tubular atrophy (IFTA), has improved throughout recent years as advances in annotation speed, predictive capacity, and breadth of utility have provided strong arguments for clinical applicability [59].

Recent studies have studied predictive AI-modeling for morphologically complex structures of the kidney using WSIs of human renal biopsy samples [62]. One such study explored the use of CNNs in semantic segmentation of glomerulosclerosis and IFTA from renal biopsies, in which assessment of CNN performance spanned three morphologic areas: IFTA, non-sclerotic glomeruli, and sclerotic glomeruli [59]. Per these respective areas, CNN demonstrated a balanced accuracy of 0.82/0.94/0.86 and MCC of 0.6/0.87/0.68 for intra-institutional holdout cases. For inter-institutional holdout cases, balanced accuracy was 0.70/0.93/0.84 with MCC of 0.49/0.79/0.64 per respective area. Investigators noted the CNN model demonstrating the best performance used a smaller network and low resolution for image analysis. In multiple cases, the CNN demonstrated the capacity to learn to predict IFTA boundaries with greater precision than the ground-truth annotations used for its training. Significant correlation was noted when comparing IFTA and glomerulosclerosis estimations via CNN with ground truth annotations, with IFTA yielding a correlation coefficient of 0.73 (95% CI [0.31, 0.91]) and glomerulosclerosis that of 0.97 (95% CI [0.9, 0.99]). No substantial difference was noted in score agreement concerning comparisons of IFTA grades as per visual assessment conducted by pathologists vs. CNN predictions against ground truth annotations, with inter-rater reliability for pathologists measured to have a kappa value of 0.69 with 95% CI [0.39, 0.99] and that of the CNN to have a kappa value of 0.66 with 95% CI [0.37, 0.96]. The CNN also demonstrated learning capacity in identifying segmental sclerosis, despite having not been trained to identify findings of this nature. Results strongly indicate the feasibility of DL-tools for high-performance segmentation of morphologically complex image structures, e.g., IFTA, by CNN.

Another CNN developed for the identification and segmentation of glomeruli on WSI of human kidney biopsies demonstrated accurate discrimination of non-glomerular images from glomerular images that were either normal or partially sclerosed (NPS) or globally sclerosed (GS) (Accuracy: 92.67% ± 2.02%, Kappa: 0.8681 ± 0.0392) [60]. The segmentation model derived from the CNN classifier demonstrated accuracy in marking GS glomeruli on test data (Matthews correlation coefficient = 0.628).

As tissue volume requirements and annotation quality often mar adoption of CNN training for quantitative analysis, investigators seeking to reduce annotation burden experimented with development of a Human AI Loop (H-AI-L), e.g., “human-in-the-loop” pipeline for WSI segmentation. Annotation speed and accuracy were noted to perform faster than traditional methods limited by data annotation speed [63].

Another ML pipeline was developed for glomerular localization in whole kidney sections for automated assessment of glomerular injury [64]. Average precision for glomerular localization was reported as 96.94%, with an average recall of 96.79%. The localizer did not demonstrate bias in identifying healthy or damaged glomeruli nor did it necessitate manual preprocessing.

Reduced variability from AI-assisted analysis of fine pathologic structures at high resolution may provide accurate quantitative assessment of WSIs for IFTA grade prediction, as demonstrated by a DL framework developed at the Ohio State University Wexner Medical Center using trichrome-stained WSIs. Strong inter-rater reliability was noted regarding IFTA grading between the pathologists and the reference estimate (κ = 0.622 ± 0.071). The accuracy of the DL model was 71.8% ± 5.3% on The Ohio State University Wexner Medical Center and 65.0% ± 4.2% on Kidney Precision Medicine Project WSI data sets (from which model performance was evaluated) [65].

The first CNN-based model relevant to kidney transplantation within the literature was developed to address significant intra- and inter-observer variability reported during donor biopsy evaluation [66]. The DL model is the first to have been developed for the identification and classification of non-sclerosed and sclerosed glomeruli in WSI of donor kidney frozen section biopsies. When trained on only 48 WSIs, the model demonstrated slide-level performance in evaluation that was noted to be on par with expert renal pathologists. The model also significantly outperformed, in both accuracy and speed, another CNN model trained using only image patches of isolated glomeruli. Investigators noted that while model training with WSI patches has demonstrated efficacy in WSI classification tasks, this is only when applied to the classification of WSI patches and did not work as effectively for WSI segmentation in the setting of their study. Authors postulated, per results achieved from this CNN model, a future in which its utilization is deemed essential for clinical evaluations of donor kidney biopsies.

A recent publication explored the development of a pipeline for the classification and segmentation of renal biopsies from patients with diabetic nephropathy [67]. The pipeline consisted of a CNN used to detect glomerular features reflective of glomerulopathic structural alteration and a Recurrent Neural Network (RNN) used for analysis of glomerular features for final diagnosis of the biopsy. The pipeline was designed to be extendable to any histologically interpreted glomerular disease, e.g., IgA nephropathy, lupus nephritis, and is trainable for the prediction of any label with a numerically associated indicator of severity such as proteinuria. Strong comparison to traditional, e.g., visual classification methods was noted. The pipeline detected glomerular boundaries from whole slide images with 0.93 ± 0.04 balanced accuracy, glomerular nuclei with 0.94 sensitivity and 0.93 specificity, and glomerular structural components with 0.95 sensitivity and 0.99 specificity. Results were congruent with ground truth classifications annotated by a senior pathologist (κ = 0.55 with a 95% confidence interval (0.50, 0.60) and two additional renal pathologists κ_1_ = 0.68, 95% interval (0.50, 0.86) and κ_2_ = 0.48, 95% interval (0.32, 0.64).

Percentage assessment for normal and sclerotic glomeruli is vital in determining renal transplant eligibility, with percentage of normal and sclerotic regions serving as, respectively, good or poor indicators for transplant outcome [68]. DL has been leveraged to improve stratification of kidney disease severity via combining patient-specific histologic images with clinical phenotypes of chronic kidney disease (CKD) stage, serum creatinine, and nephrotic-range proteinuria at time of biopsy and afterward [69]. CNN models were demonstrated to outperform score assessments for pathological fibrosis undertaken by pathologists for all clinical CKD phenotypes. In comparison to pathologist estimation, CNN prediction for CKD stage yielded greater accuracy (κ = 0.519 vs. 0.051). CNN demonstrated an AUC of 0.912 vs. an AUC of 0.840 measured for pathologist estimations for creatinine. For proteinuria estimation, CNN AUC was 0.867 vs. 0.702. CNN estimations for 1-, 3-, and 5-year renal survival yielded respective AUC values of 0.878, 0.875, and 0.904 vs. 0.811, 0.800, and 0.786 via pathologist assessment.

Histopathological images are ripe with information exploitable for clinical survival and therapy response prediction. Such information may be buttressed with supplementation of categorical pathology-report data, as indicated in the previous examples. Histopathological data typically analyzed from WSIs for the prediction of survival and therapy response may also be effectively supplemented with pathology images from multiple sources, as demonstrated in a recent study evaluating an AI-pipeline developed for the prediction of neoadjuvant chemotherapy (NAC) response for patients with breast cancer. 

### 3.3. Optimizing Machine Learning for Neoadjuvant Chemotherapy Response 

Immunohistochemistry (IHC) WSIs are replete with data that may be utilized as a powerful adjunctive to histopathology WSIs. IHC images may be quantified for biomarker results, e.g., PD-L1, ER, PR, HER2, Ki67, and distribution of biomarker expression, e.g., PD-L1 (tumor and inflammatory cells), CD8 (cytotoxic tumor-infiltrating lymphocytes/TILs), and CD163 (type 2 macrophages), both metrics of which are important in predicting tumor response to chemotherapy. 

#### 3.3.1. Modeling Predictive Response for Neoadjuvant Chemotherapy in Breast Cancer

Up to 50% of HER2-positive breast cancers and a subset of triple-negative breast cancers (TNBCs) achieve pathologic complete response (pCR) following neoadjuvant chemotherapy (NAC), thereby allowing NAC response to act as corollary for disease-free survival in TNBC and HER2+ breast cancer patients [70,71,72]. Many factors are associated with pCR in breast cancer, e.g., higher mitotic activity and tumor (Nottingham) grade are associated with higher frequency of pCR [73]. Tumor-associated lymphocytes (TIL) occur with greater frequency in TNBC and HER2+ breast cancer subtypes [74]. PD-L1 expression, particularly in HER2+ patients, has demonstrated association with pCR in breast cancer [75,76]. Hormone receptor level is also associated with pCR, with ER-/PR-/HER2+ breast cancers demonstrating the greatest likelihood for pCR amongst all HER2+ tumors [77]. Higher intensity of HER2 IHC expression is associated with significantly higher likelihood for pCR in HER2+ breast cancer than for cases with incomplete pathological response [78]. Intratumoral heterogeneity is independently associated with incomplete response to anti-HER2 NAC in HER2+ breast cancer.

A recent groundbreaking effort compiled multiple image-based features extracted from multiple sources, i.e., H&E-stained WSIs and IHCs (PD-L1, CD8, CD163), quantitative and qualitative breast cancer biomarker results (ER, PR, HER2), and patient demographic and clinical features, e.g., age, to develop a predictive ML model for NAC response in TNBCs and HER2+ breast cancers. An automatic WSI feature extraction pipeline in which H&E-stained WSI tissue segmentation utilized a well-trained neural network model (DeepLabV2) to generate stromal, tumor, and aggregated lymphocyte areas (distinguished by computerized colorization). Multiplexed IHC WSI (CD8, CD163, PD-L1) segmentation was performed using color-based K-means segmentation, in which entire WSIs were segmented into three different IHC areas, then followed by an automatic, multi-step, and non-rigid (changing image size, but not shape) histological image alignment (“registration”) of H&E and IHC, upon which an algorithm selected the best non-rigid transformations. Three categories of quantitative IHC image features (CD8, CD163, PD-L1) were extracted from the registered WSIs for subsequent evaluation of expression and distribution within different cellular components/regions (stroma, tumor, lymph) including an overall evaluation of all tissue components. Area ratio, proportion, and purity of IHC image features within cellular regions was evaluated. Breast biomarker results, e.g., positivity/negativity, percentage, were evaluated within inclusion of additional demographic characteristics in relation to different IHC markers, with data pooled from individual and combined H&E/IHC sources. 

The ML model predicted NAC outcomes using the various extracted image features using a form of logistic regression. Four groups of image features were compared using AUC, F-1 score, precision, and recall measurements for HER2+ and TNBC patient cohorts:All pipeline-extracted features (36 total) and clinical data patient features, e.g., including biomarker results, age, and additional demographic factors)Automated/pipeline-extracted H&E-stained WSI and clinical patient data featuresAutomated/pipeline-extracted IHC WSI and clinical patient data featuresPathologist-extracted WSI features and clinical patient data features

Algorithmic models were developed per each group of pipeline-extracted WSI features/clinical features (with the ML model from the fourth group trained using manually extracted features by pathologists). For both the HER2+ and TNBC cohort, the first group performed best in each measurement, especially for the HER2+ cohort. A feature importance analysis was conducted in which favorable and unfavorable features predictive of pCR or residual tumor, respectively, were determined for both patient cohorts. Favorable features for the HER2+ cohort were determined by the ML model as the independent ratios of CD8, CD163, and PD-L1 in the lymphocytic region, CD163 ration in the tumor area, and the HER2/CEP17 ratio. Unfavorable features for the HER2+ cohort included age, ER and PR ratios, PR positivity, and the stromal CD8 proportion. Overall results demonstrated the effective capacity of the AI-pipeline to automatically extract H&E and IHC image features with accuracy. ML models developed based upon the pipeline-extracted WSI features and clinical features demonstrated the potential for NAC response prediction in breast cancer patients while outperforming the algorithm trained by pathologist-extracted features. The AI-pipeline also generated image features that could be used to predict residual cancer burden in breast cancer cases with residual tumor. 

#### 3.3.2. ML for Subspecialty Practice Survival Modeling

Tabibu et al. [34] provided encouraging data following the development of a CNN for the automated subtype classification of renal cell carcinoma (RCC) and identification of features predictive of patient survival outcome. A total of 1027 ccRCC, 303 Papillary RCC, and 254 Chromophobe WSIs with corresponding clinical information were selected for model training from the Cancer Genome Atlas, with 379, 47, and 83 normal tissue images per each respective RCC subtype. An accuracy of 99.39% and 87.34% was recorded for classification of ccRCC from normal tissue and chromophobe RCC from normal tissue, respectively. The AI-model classified ccRCC, chromophobe, and papillary RCC with 94.07% accuracy. High-probability tumor regions identified by the CNN were targeted for morphological feature extraction used for prediction of ccRCC patient survival outcome. Significant association with patient survival was found after generated risk index was derived based upon tumor shape and nuclei from the extracted regions. 

Prediction of RCC recurrence following nephrectomy has also seen focus for ML development, as outlined in a recent study assessing recurrence probability 5- and 10-years post-nephrectomy. Analytical data from 2814 RCC patients were used for model testing, which yielded AUC values of 0.836 and 0.784 5- and 10-years following nephrectomy [79]. 

## 4. Actuating Clinical Implementation through Achieving Generalizability

The essence of generalizable AI for clinical pathology lies within the capacity for an AI-tool to remain robust in its precision, accuracy, and efficiency in executing a diagnostic function when confronted with a broad range of tissue variations potentially encountered within a daily clinical workload. 

Small, localized cohorts, insufficient ground-truth determination from expert pathologists, non-standardization of training methods and materials and lack of external validation are only some of many risks which have hampered the clinical generalizability of studies that have otherwise presented highly encouraging data. Circumvention of this key barrier to achieving deployment of AI within clinical practice requires equal applicability of an ML tool to different patient populations, pathology labs, WSI scanning device models, and reference standards derived from intercontinental specialist pathologist panels [80].

The largest collective effort for generalizable AI for prostate cancer diagnostics was reached during the Prostate Cancer Grade Assessment (PANDA) challenge, in which 12,625 prostate biopsy WSIs sourced from six international sites were used for model-development, performance evaluation, internal, and external validation [80]. Histological preparation and scanning of WSI data used for external validation was performed by multiple independent laboratories and was compared to pathologist reviews. On United States and European external validation sets, the algorithms achieved agreements of 0.862 (quadratically weighted κ, 95% confidence interval (CI), 0.840–0.884) and 0.868 (95% CI, 0.835–0.900) with expert uropathologists [80].

Well documented accounts of AI-model development for pathology during the last two years have involved large numbers of patient cases for training, testing, and validation data sets, interpretations by multiple expert pathologists to establish ‘ground truth’ for diagnosis, use of slides from multiple institutions, and use of differing scanners including scanners from external institutions [23,26,28,30,81]. 

WSIs of 12,132 prostate needle core biopsies digitized by two different WSI device models at Memorial Sloan Kettering (MSK) were used to train a DL system that was tested on 12,727 prostate needle core biopsies from institutions around the world [23]. Investigators found that approximately 10,000 slides were necessary for effective training of their system [82]. Authors noted a 3% difference in AUC recorded between WSI devices used for image capture and digitization attributed to variations in brightness, contrast, and sharpness between the devices. Investigators postulated that the AI-model could remove >75% of slides from a standard pathologist workload without compromising sensitivity and facilitate an increased user-base of non-subspecialized (non-GU pathologists) who may diagnose prostate cancer with greater confidence and efficacy when aided by the algorithmic tool. Weakly supervised AI-model training linking every WSI to synoptic data elements, e.g., benign vs. adenocarcinoma, provided a scalable mechanism of dataset creation circumventing data limitations which often mar the capacity and clinical implementation of highly supervised DL algorithms. Through using only label-based diagnoses for training WSIs, investigators were able to eschew any form of labor-intensive and time-consuming data curation including pixel-wise manual annotations used in highly supervised model training.

High-volume model training using 36,644 WSIs, 7514 of which had cancerous foci, was used in early development of diagnostic software for prostate adenocarcinoma recently granted de novo marketing authorization for in vitro diagnostic (IVD) use, signifying the first ever FDA-approved AI product for clinical pathology [83]. A total of 304 expertly-annotated prostate CNB WSIs were used to establish ground truth for evaluation of the DL system (Paige Prostate Alpha^®^, Paige AI, New York, NY, USA) [27]. An average diagnostic sensitivity of 74% and specificity of 97% was recorded for general pathologists prior to use of the DL system. When aided by the AI-tool, sensitivity increased to 90% while specificity remained the same. Results suggested the utility of the tool for second-read applications, e.g., quality assurance. Such a device could be deployed in settings where GU pathology subspecialists are not commonly, if at all, present, e.g., underserved climates with substantial healthcare disparity. 

The Paige Prostate^®^ system, successor to prototypal version Paige Prostate Alpha^®^, was subject to extensive multinational validation spanning ≥150 different institutions and a diversity of clinical and demographic characteristics from ≥7000 patients including differing tumor sizes, grades, and patient ethnicities [23,83,84,85,86,87]. The system achieved a sensitivity of 97.7% and a specificity of 99.3% in detecting cancer in 1876 prostate CNB WSIs, also demonstrating 99% sensitivity and 93% specificity at part-specimen level while upgrading pathologist-ascribed benign/suspicious patient diagnosis to malignant after identification [84,85].

The strengths of many high-volume studies for the validation of Paige Prostate^®^ and other software systems which have since seen clinical deployment throughout the globe now may serve as guidelines for appropriate model evaluation for clinical generalizability. Large cohort sizes, testing sections containing substantial pre-analytic artifacts, e.g., thick cuts, fragmentation, and poor staining, abundance of challenging histological patterns including those seen in benign-mimicking malignant prostatic adenocarcinoma, e.g., pseudo hyperplastic and atrophic pattern variants, along with benign histology that may be mistaken for prostatic adenocarcinoma, all are variables which may confound the appropriate detection and grading of prostatic adenocarcinoma for an insufficiently trained AI-model, yet did not pose hurdles for the DL-models that would later see clinical implementation [88,89,90].

### Clinical Integation of AI

AI tools have demonstrated real-world merit for quality control (QC) support and first read applications for primary diagnostic use within clinical settings. The Paige Prostate^®^ solution notably reduced time-to-diagnosis by 65% when applied to diagnostic histopathologic data from 682 TRUS prostate needle biopsy WSIs acquired from 100 consecutive patients at a laboratory unassociated with its original development and validation [87]. The AI-system notably demonstrated 100% sensitivity and negative predictive values for patient-level diagnostics.

CorePlus (CorePlus Servicios Clínicos y Patológicos LLC, a high complexity CLIA-certified clinical and anatomic pathology laboratory is the first U.S. laboratory to integrate an AI-platform for diagnostics, lab efficiency, and quality control [91]. The Galen™Prostate solution (CE-marked; Ibex Medical Analytics) was integrated into the fully digitized laboratory for routine clinical second-read diagnostic applications. The AI-solution was previously clinically validated for routine clinical diagnostics involving detection, grading, and evaluation of clinically relevant findings within WSIs of prostate CNBs in an extensive study demonstrating the utility of the AI-solution for routine clinical practice [28].

The study was the first to evaluate the performance of a prostate histopathology algorithm deployed within routine clinical practice for assessment of cancer detection, Gleason grading (GG), and proportion of tumor extent in addition to detection of perineural invasion, demonstrating the multifaceted merits of the AI-solution which may fulfil a gamut of clinical reporting needs. Algorithmic interpretation of perineural invasion (PNI) within CNB WSIs, a typically small and relatively uncommon finding bearing large clinical and prognostic significance, presented unique focus for investigators as previous studies had not reported AI-based detection for the feature. The algorithm’s capacity to simultaneously evaluate CNBs for PNI (AUC: 0.96 external validation dataset) while interpreting a battery of standard metrics for prostate CNB, e.g., cancer detection, grading, and tumor extent, highlighted the ability of the AI-platform to execute a multitude of functions with high performance. 

The second-read application of the AI-platform was again clinically validated for a unique patient population bearing high rates of prostate cancer-specific mortality at the CorePlus laboratory and assessed via comparison to pathologists diagnoses for ground truth, yielding encouraging results including accurate identification of benign vs. cancerous tissue (AUC: 0.994; Specificity: 96.9%; Sensitivity; 96.5%) and GG 1 vs. GG 2+ (AUC: 0.901; Specificity: 81.1%; Sensitivity: 82.0%). 

Following clinical implementation of the Galen™Prostate solution at the CorePlus laboratory for primary application in QC, the AI-tool has discovered and corrected 1.97% of over 4000 cases (encompassing over 54,000 WSIs since deployment of the AI platform in June 2020) incorrectly identified as false-negatives. During this period, the second-read application identified 51.4% of cases as benign, 18.16% as GG1, 29.83% as GG2+, while providing technical alert notifications for 1.79% of WSIs. In total, 100% of PCNBs at the laboratory are analyzed with the assistance of AI prior to sign-out.

The Galen™ platform (including Galen™Prostate and Galen™Breast, CE-marked for clinical breast cancer diagnostics) was also integrated into the clinical workflow for second-read applications at Maccabi Healthcare Services (Israel), a large, centralized pathology institute that receives samples from 350 surrounding clinics and hospitals. A significant proportion of yearly histopathology workload at the institution consists of PCNBs (700 cases per year; >8000 slides). Alerts from the second-read application, viewable from the case list and outlined by heatmaps displayed in a slide-viewing module, were raised in 10.1% of PCNB WSIs (583) taken from 232 cases initially given benign diagnoses by pathologists. Gleason 7+ alerts were raised in 5.3% of slides (93) taken from 137 cases initially given diagnoses of Gleason grade 3 + 3. Alerts from the AI-system significantly streamlined the review process and required minimal review time from pathologists (approximately 1% of FTE).

The value of AI-assisted QC was again demonstrated in an earlier study assessing the performance of the Galen™ Prostate algorithm after its validation at Maccabi [92]. Results: Following deployment in four laboratories within the Medipath network, the largest system of pathology institutes in France (averaging an annual workload of 5000 PCNB cases), the AI-solution was noted to have identified 12 cases misdiagnosed as benign, some of which the system identified as having high-grade cancer.

AI-assistance for first read applications offered by the Galen™Prostate solution underwent clinical validation in which superior outcomes were yielded from pathologists using the AI-tool in comparison to those using only light microscopy during interpretation of 100 PCNBs. Results demonstrate a 32% reduction in major discrepancy rate with use of the first read application [93].

Improvements in productivity, clinical-grade accuracy, TAT, and case-level discrepancy resolution were observed after clinical integration of the Galen™first read AI-system at Maccabi Healthcare Services. A 27% overall reduction in time-to-diagnosis and 37% overall gain in productivity compared to manual microscopy followed deployment of the AI-tool, which yielded a 32% reduction in time-to-diagnosis for benign cases and a 25%-time reduction for those with prostate cancer [94]. Diagnostic accuracy did not suffer from increased efficiency, as results for case-level diagnostic accuracy were congruent between AI and manual microscopy. A similar trend was also observed for resolution of case-level discrepancies, of which the Galen algorithm was able to deliver a 97.8% agreement with ground truth following discrepancy resolution in comparison to the almost equivalent 97.5% for diagnosis via microscopy [94]. In total, 160 cases (1224 slides) were used to evaluate case-level agreement for primary diagnosis. A 95.3% agreement was noted between the AI-solution and microscopy diagnosis for 378 cancerous and 789 benign WSIs in a study evaluating the performance of the AI-solution against 310 PCNB cases (totaling 2411 H&E slides) [94]. A total of 99.7% of pathologists using AI-assistance in the study agreed with classifications provided by the adjunctive tool, including reclassifications of three false-negative slides initially classified as benign. It was observed that the use of AI did not yield any false negative diagnosis throughout the duration of the study. Examples of cases misdiagnosed via manual microscopy and detected with AI included high-grade prostate adenocarcinoma (GG4 + GG3) and a case in which only one slide demonstrated findings of prostate cancer [94].

Turnaround time, e.g., total time from first to last review and sign out of one patient case, was also demonstrably reduced with AI-assistance, markedly reducing TAT by 1–2 days while enabling a single review for almost all cases. A total of 80% of cases analyzed via standard microscopy included additional ordering of IHC. Only 0.6% of cases interpreted via AI involved additional IHC ordering to those already automatically pre-ordered based upon AI-classification. In evaluation of 238 cases interpreted via microscopy vs. AI-assistance, the mean sign out time for pathologists using standard microscopy was reported as 1.8 days in comparison to 9.4 min via AI.

## 5. Discussion

DP technology has revolutionized laboratory practices through enabling the digitization and viewing of entire laboratory histopathological glass slide workloads at microscope resolution. Four generations of WSI scanning instruments have passed since the 2001 release of the Leica Aperio T1, with each generation marking successive improvements in scanning speed, image quality, and batch scanning capacity. Yet, although mature technology to support WSI and laboratory digitization is now readily available, supportive of high-volume laboratory integration, and more cost-effective for implementation than ever before, few laboratories to date have undergone complete digital transformation for routine clinical practice [11,95]. Commercial AI-tools for diagnostic pathology are primed for adoption, offering “plug-and-play”, user-friendly systems which now include applications for case triaging, worklists, slide viewing, IHC pre-ordering, tumor grading, sample measurements, reporting, and identification of non-cancerous findings. An increasing number of vendors offer dedicated software with algorithms for WSI image analysis, e.g., estrogen receptor, progesterone receptor, and human epidermal growth factor receptor 2 scoring, including automated multi-class segmentation of H&E stained WSIs demonstrated via ‘heatmapping,’ i.e., colorized pixel wise classification of tissue (Figure 11) [96].

Commercial vendors are now utilizing modern DL techniques including multiple instance learning (MIL) to generate cell-by-cell data, quantify within subregions, and perform feature-based analysis for multiple applications in brightfield and fluorescent WSIs, with recent solutions facilitating the accurate prediction of genomic status from H&E-stained slides [97]. An increasing number of commercially available AI solutions support the construction of bespoke AI models, e.g., custom assay development, whereby pathologists are enabled to train models via annotations rather than complex coding, with some vendors offering built-in learning tools to further assist the annotation process for faster training and increased accuracy. Commercial solutions are capable of high throughput with fast turnaround from rapid generation, visualization, and export of spatial, morphological, and increasingly precise histology data from WSI, e.g., tumor area, cell counts, cell size, staining intensity, collagen area, and blood vessel density [98]. Yet, only 22% of laboratories enrolled in a 2016 College of American Pathologists (CAP) quality improvement survey for histology reported using quantitative tools for image analysis tasks [99].

Following the introduction of IHC and next generation sequencing for clinical practice, AI has been deemed the third revolution of pathology [99]. Such a revolution and the magnificent potential it holds for clinical pathology, as evidenced in our review, may only be unleashed through WSI adoption. Improvements in clinical-grade accuracy, increased productivity, robust QC, and shorter TAT realized through AI-augmented DP laboratories have driven a case for full laboratory digitization as seen in Maccabi Healthcare Services. Integration of AI into clinical workflow at the healthcare network yielded low risks while producing high returns. These findings, in conjunction with increased efficiency, decreased IHC ordering, and increased practitioner satisfaction supported the decision for complete digitization from both practical and financial vantage points. 

Van Der Poel et al. had noted in their 1992 review of computational applications in “quantitative light microscopy” for the diagnosis of prostatic adenocarcinoma, transitional cell carcinoma, and renal cell carcinoma, that the purpose for investigating the application of such techniques stemmed from the highly inconsistent nature of visual tumor grading lending to high interobserver variability reported at the time [16]. Authors also noted changes in grading systems that were often descriptively subjective therefore resulting in “disturbingly low reproducibility”, further compounding the highly subjective nature of pathologist-directed quantification of histological, cellular, and nuclear features pertaining to malignant potential. The comprehensive review of primordial computational applications for GU pathology concluded with noting that such techniques were valuable in aiding diagnosis only when confined to research settings. The need for standardized automated fixation, embedding, staining, selection, and measuring techniques was emphasized, as the extensive data analyzed within the review had been obtained with varying preparation methods and therefore differed too greatly to support any consistent conclusions.

Twenty years later, Egevad et al. reported on their investigation concerning the shifting approach to Gleason grading following the 2005 change in guidelines by the International Society of Urological Pathologists [36]. New encouragement to incorporate poorly formed glands and cribriform patterns into Gleason pattern (GP) 4 had led to high inter-observer variability amongst even specialist urologic pathologists (κ = 0.34), who expressed concern regarding the compromised significance of GS 7 in the wake of the amended guidelines [35].

Modern computational pathology tools have facilitated the standardization of workflow components highlighted by Van Der Poel et al., yet most current studies involving the interrogation of AI-development for pathology are still relegated to academia while lacking any consistent methodical standardization that may be utilized for clinical relevancy. Though the performance of AI algorithms for GU pathology has, within the cohort of research studies included in our review, demonstrated equivalency to the diagnostic interpretations of GU specialists while surpassing those of general pathologists, training materials and methods for individual AI-models varied when evaluated as a conglomerate. Variations in tissue samples and WSI patch sizes used for model training are two such examples. Firm conclusions pertaining to the clinical relevancy of these investigations were also unreachable, in some instances, due to lack of model-development for correlating cancer grade with clinical outcome. 

In his wake, Aurelius left scores of stoic prose directing those seeking understanding almost two millennia later to “look back over the past, with its changing empires that rose and fell…” to foresee the future. Those pursuing additional directives towards fomenting solutions for change may turn to Socratic texts encouraging avoidance of conflict with the old to build upon the new, words which predated Aurelius’ reign by centuries.

Realization of objective diagnostic reproducibility has been a coveted goal for clinical pathology long before the concept of computer vision, or computers in any respect. Newer studies featured in our review have highlighted instances of clinical implementation, groundbreaking generalizability, and MIL methods for algorithm development incorporating data from entire clinical pathology reports into training for enhanced clinical relevancy. Newer CNN derivatives and methods for model training have emerged to combat data concerns which have been a primary limitation to algorithmic clinical implementation. 

Integration of a fully digitized, LIS-centric laboratory was a response to overwhelming workload burdens at the Gravina Hospital in Caltagirone (Sicily, Italy) [95]. Such problems had been compounded by the Coronavirus pandemic, though were omnipresent in similar departments throughout the globe struggling to combat long withstanding problems that had fervently resurged after lying dormant for years. Digital transformation of all workflow steps through departmental LIS allowed practitioners and staff at the Gravina Hospital to alleviate burdens through a completely interconnected, easily streamlined workflow. DP transformation of the laboratory would later be followed by augmentation of the digitized workflow processes with AI-tools. Superior diagnostic concordance amongst pathologists and increased WSI quality was observed shortly following implementation of AI, as algorithmic adjuncts to digital workflow processes created a cause for upholding a high standard of workflow quality. Through shining a light upon areas for improvement within the existing workflow it was embedded within, AI had uncovered problem areas otherwise overlooked existing prior to its arrival, which would then be amended to optimize the conjunctive potential of both diagnostic utilities.

Unearthed depths of clinical potential embedded within millions of WSI pixels has driven the development of effective, accurate, and precise AI algorithms purposed for transforming such potential into meaning. The same prospects have inspired the evolution of four generations of WSI devices to extract and present data from glass slides with greater efficacy, accuracy, and precision with each successive iteration. Driving forward and fueled through shared inspirations of clinical prospect and potential, WSI has met AI at a road converged where both may continue to a destination of enhanced clinical understanding and optimized patient care. As the night turns and AI embeds itself within the digital bedrock of clinical pathology, Rushmore-esque pillars form to cast monumental gazes of computer vision to an infinitely opportune landscape ahead.

## Figures and Tables

**Figure 1 diagnostics-12-01778-f001:**
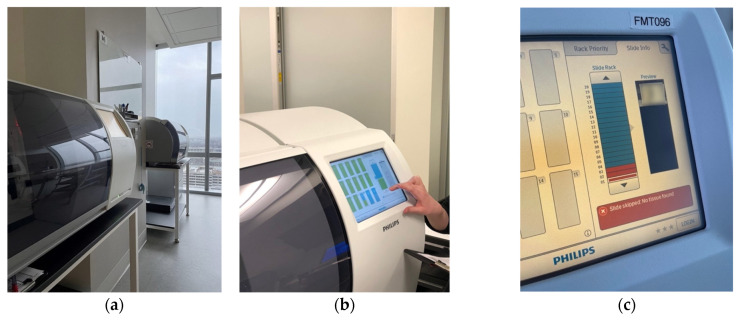
Digital pathology integration at the Ohio State University James Cancer Hospital and Solove Research Institute (captured by David Kellough of The Ohio State University Comprehensive Cancer Center—Arthur G. James Cancer Hospital and Richard J. Solove Research Institute): (**a**) Philips UFS scanners; (**b**) Integrated LCD touchscreen for WSI review; (**c**) Scan failure indicator; (**d**) example of scanning error (“Venetian blinding”); (**e**) Histology laboratory.

**Figure 2 diagnostics-12-01778-f002:**
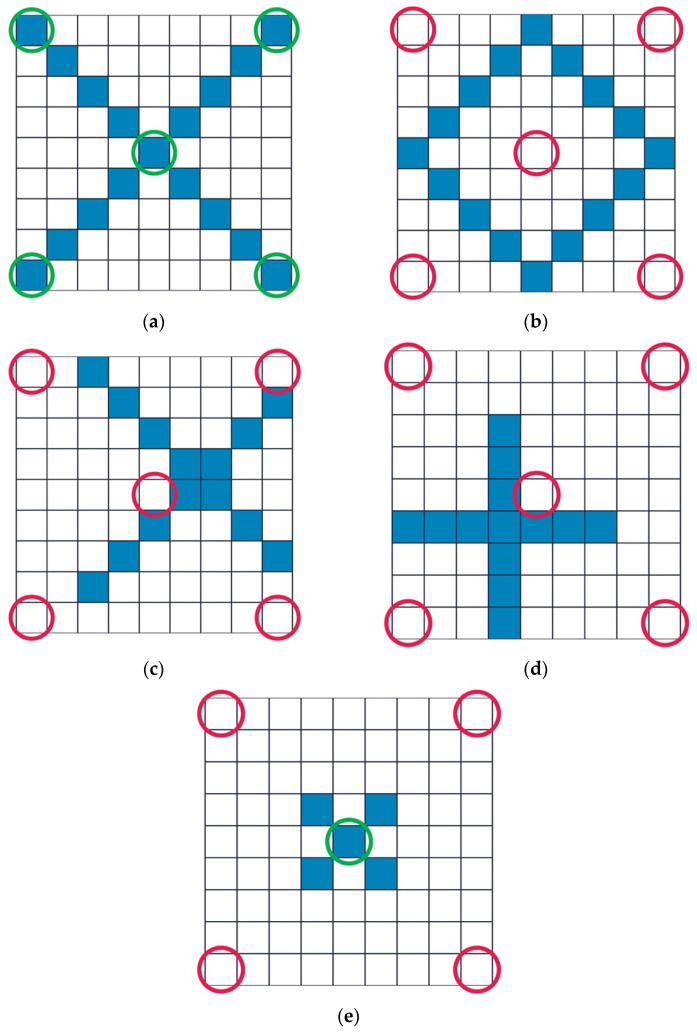
Traditional Programming vs. Machine Learning for Computer Vision (original figures). Squares are representative of pixels comprised of binary graphical indicators for computer recognition, with blue squares comprising a pixelated “input” shape to be recognized by a preset formula that may direct the computer to correctly identifying the shape in its output determination. Green circles are indicative of computer-recognized elements of the pixelated input shape per input programming rules. Red circles represent areas in which programming rules neglected to recognize blue input image elements. Computer programmed rules for defining shapes in figures (**a**) through (**e**) are (1) shape is “X” if the center and corner pixels are full and “O” if the center and corner pixels are empty: (**a**) Pathologist/human interpretation of image: shape is “X”. Computer interpretation of image: shape is “X”, as dictated by rule. Outcome: concordant with pathologist visual interpretation; (**b**) Pathologist interpretation of image: shape is “O”. Computer interpretation of image: Shape is “O”, as dictated by rule. Outcome: concordant with pathologist visual interpretation; (**c**,**d**) Pathologist interpretation of image: shape is “X”. Computer interpretation of images: shape is “O”, as dictated by rules. Outcome: discordant with pathologist visual interpretation; (**e**) Pathologist interpretation of image: shape is “X”. Computer interpretation of image: image is not recognized, as complete criteria are not fulfilled for either rule. Outcome: discordant with pathologist visual interpretation, i.e., unidentifiable image.

**Figure 3 diagnostics-12-01778-f003:**
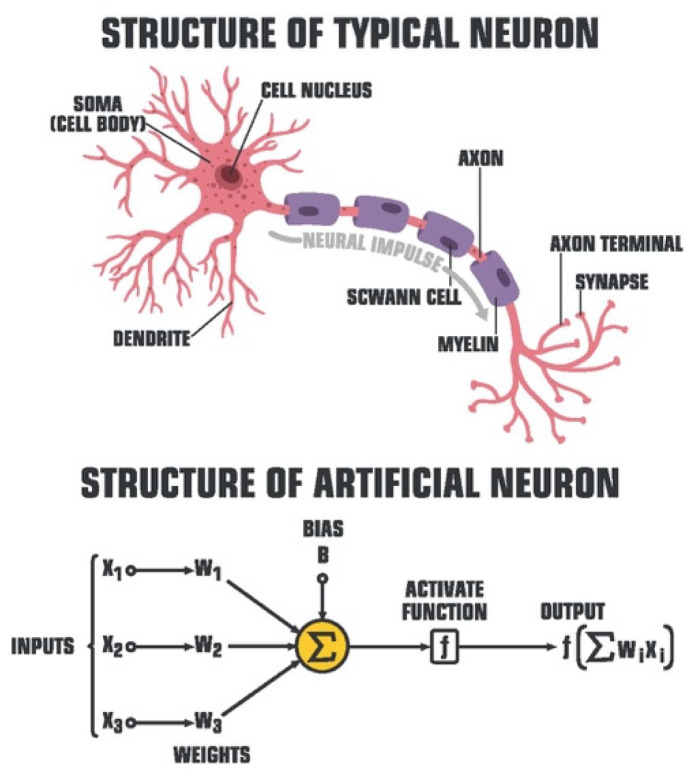
Artificial Neuron Model (ShadeDesign/Shutterstock.com, accessed on 27 April 2022).

**Figure 4 diagnostics-12-01778-f004:**
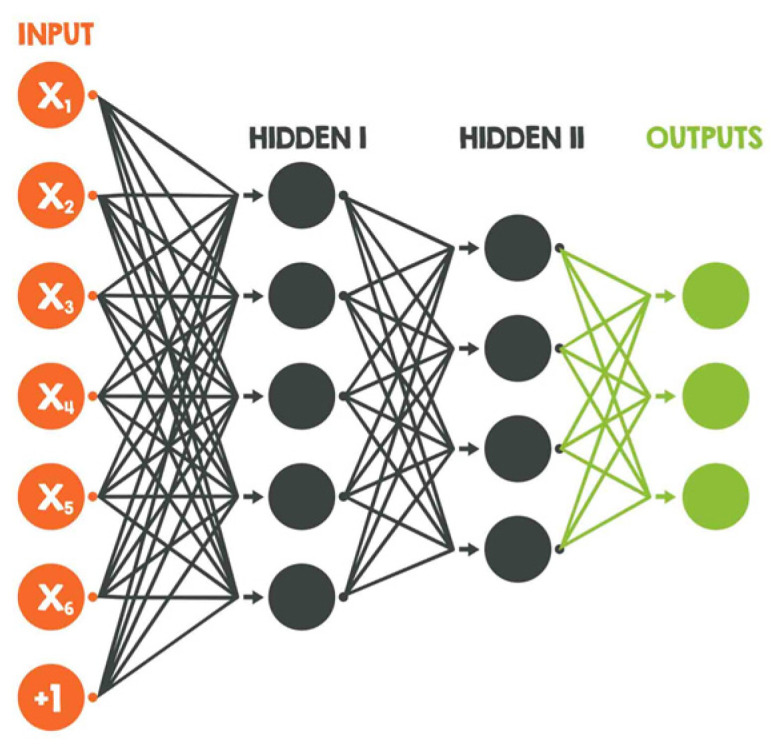
Artificial Neural Network (ShadeDesign/Shutterstock.com, accessed on 27 April 2022).

**Figure 5 diagnostics-12-01778-f005:**
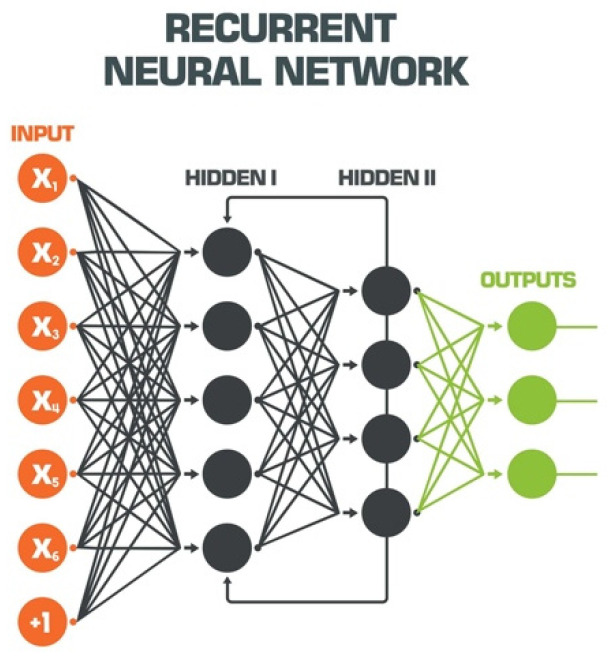
Recurrent Neural Network (ShadeDesign/Shutterstock.com, accessed on 27 April 2022).

**Figure 6 diagnostics-12-01778-f006:**
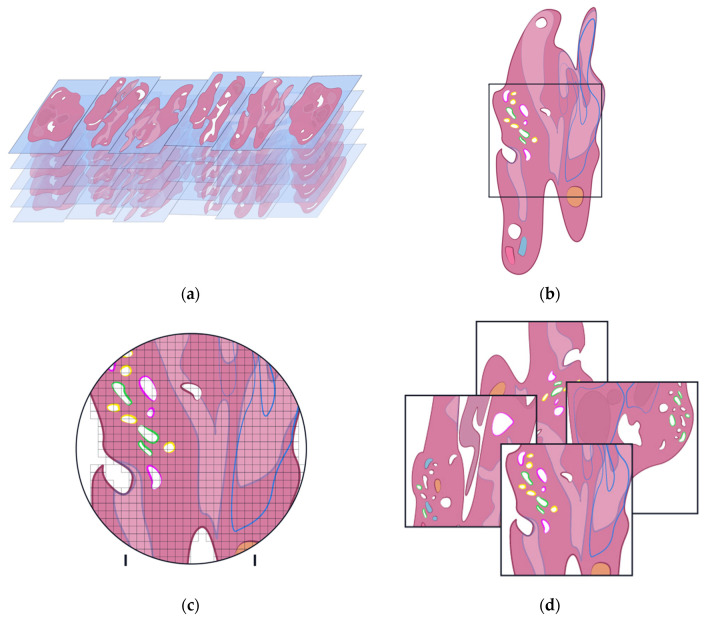
WSI patch extraction for algorithm training (original figures). (**a**) WSI dataset; (**b**) WSI region of interest for patch selection; (**c**) patch selection; (**d**) extracted patches used for algorithm training. Demarcated and colored areas on pink WSI specimen represent computer-assisted pathologist annotations.

**Figure 7 diagnostics-12-01778-f007:**
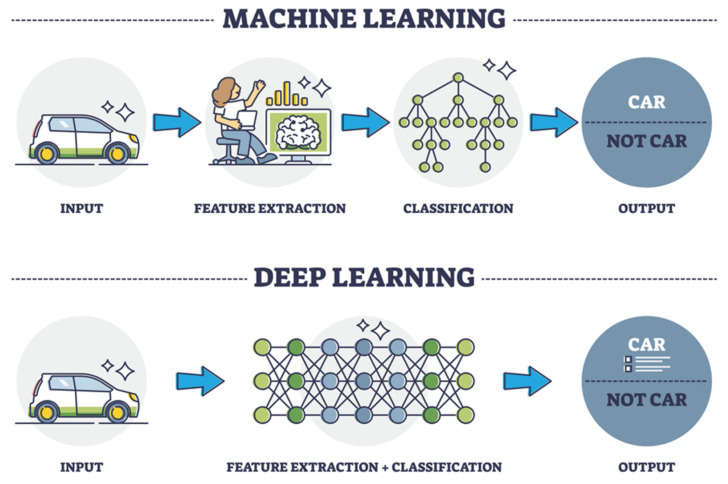
Machine learning vs. Deep learning (VectorMine/Shutterstock.com, accessed on 27 April 2022).

**Figure 8 diagnostics-12-01778-f008:**
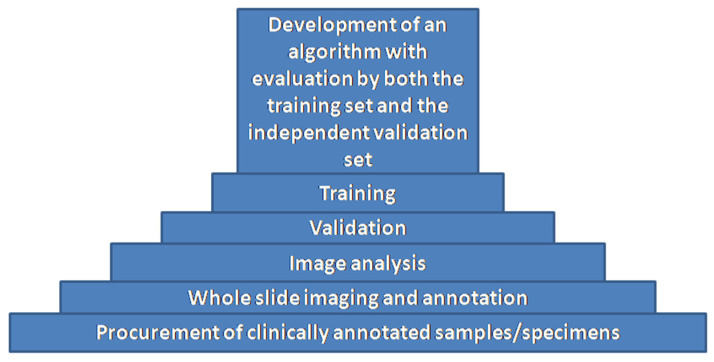
Algorithm Development (original diagram).

**Figure 9 diagnostics-12-01778-f009:**
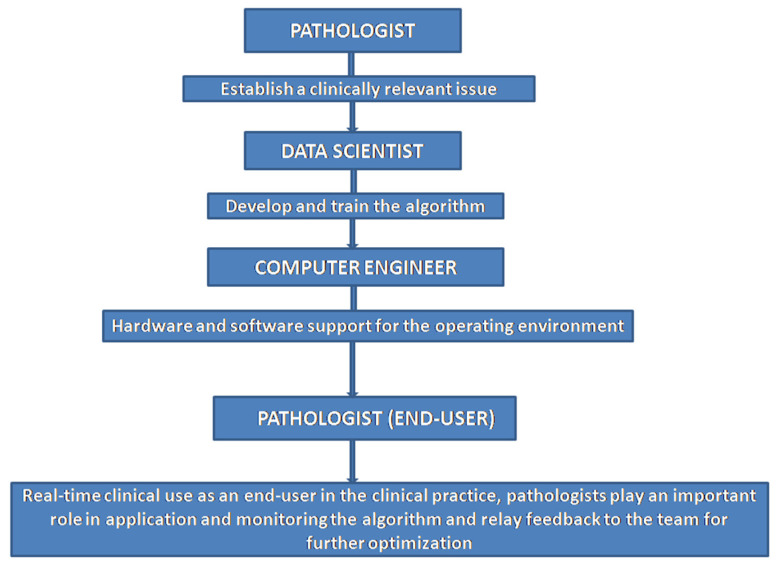
Computational pathology team (original diagram).

**Figure 10 diagnostics-12-01778-f010:**
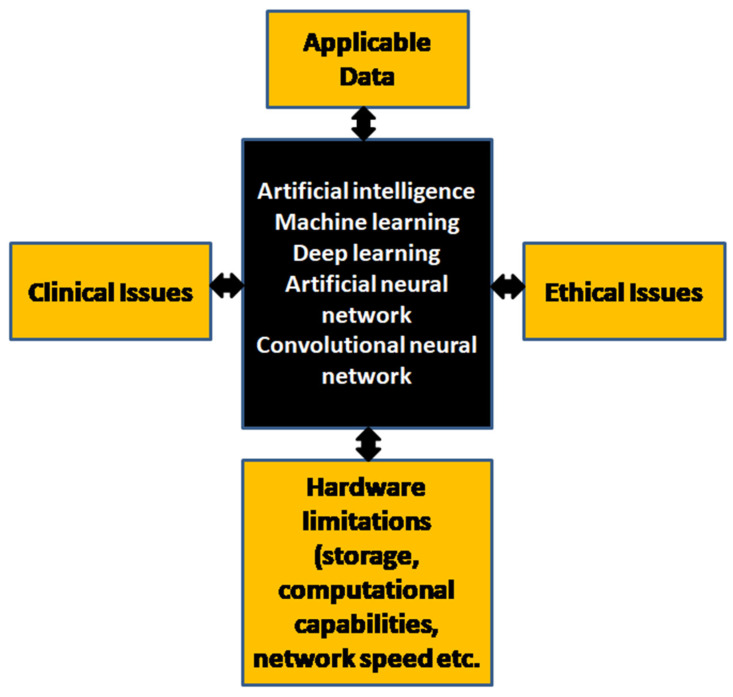
Limitations affecting AI development and deployment (original diagram).

**Figure 11 diagnostics-12-01778-f011:**
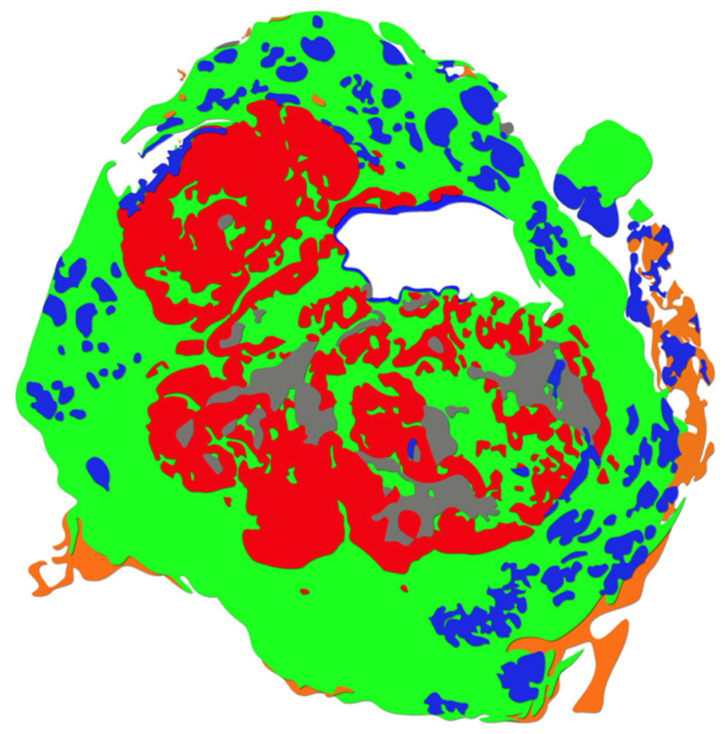
Heatmapping (original figure). Example of heatmapping in which spatial information is delineated by colors indicating carcinomatous region (red), benign epithelium (blue), stroma (green), adipose (orange), and areas of necrosis (gray) on a WSI specimen.

## Data Availability

Not applicable.

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
