# Peer review of "Cultivating Clinical Clarity through Computer Vision: A Current Perspective on Whole Slide Imaging and Artificial Intelligence"

_diagnostics, 2022, doi:10.3390/diagnostics12081778_

Round 1

Reviewer 1 Report

- The authors provided a review on the application of Artificial Intelligence (AI) on whole slide imaging (WSI) (for problems such as tumor type and stage classification).

- As a review paper, it is informative and engages with the user via fluent language.

- It could benefit from comparative tables (with work name and the metrics) for different problems it focuses on, such as (1) automated detection of cribriform pattern and (2) Gleason grade identification. That would enable the reader to grasp the strengths and weaknesses of all the referred work in an efficient manner.  

Author Response

Dear Esteemed Reviewer,

Thank you kindly for your time in reviewing this extensive manuscript, for your excellent suggestions, and for your guidance.  We absolutely agree that this manuscript would certainly benefit from reader-friendly tables inclusive of side-by-side comparative metrics from associated studies for easier digestion of the litany of diagnostic foci we have decided to incorporate within this text.  Although attempts were made to construct tables in concordance with your suggestions, of which I felt highly valuable, we were unable to incorporate these additions into the manuscript given the allotted window for amendments in conjunction with preexisting time constraints.  As the metrics of studies included within our review are categorically variable, this presented additional difficulties in presenting data in a holistically clear and concise format.

Please see the attachment ("diagnostics revisions") with suggested amendments included in:

- Figure 1 (lines 83-84), Figure 3 (line 214), Figure 4 (line 221), Figure 5 (line 237), Figure 7 (line 258)

- Figures 2, 6, and 8-11

And from requested additional content included in lines 808-830.

We hope that you may understand and feel your advice to be invaluable as it provides guidance for effective data presentation extendable to all manuscripts we propose and partake in henceforth, and will be of utmost importance for inclusion during initial draftings so that we may home data presentability in all formats, e.g., tables, figures, text, for reader satisfaction, prior to submission.

Reviewer 2 Report

Patel et al. presented an extensive review of "Cultivating Clinical Clarity through Computer Vision: A Current Perspective on Whole Slide Imaging and Artificial Intelligence". In this review, applications of machine learning and artificial intelligence in digital pathology and whole slide imaging are presented. Starting with the historical perspective, implementations of machine learning in whole slide imaging are discussed. Several examples of such implementations are reviewed as well.  The review is well-prepared and includes a fairly good discussion of the state of the art.  I have the following comments:

Figures and data taken from elsewhere should be indicated in the caption with an appropriate statement of copyright. Such as Figure 1, Figure 3, Figure 6, etc. 

Future perspective could be extended by giving more examples of the commercial applications of ML and AI in whole slide imaging.

Author Response

Dear Esteemed Reviewer,

Thank you kindly for your time in reviewing this extensive manuscript, for your excellent suggestions, and for your guidance.

Please see the attachment ("diagnostics revisions") with suggested manuscript amendments and additions seen in:

- Figure 1 (lines 83-84), Figure 3 (line 214), Figure 4 (line 221), Figure 5 (line 237), Figure 7 (line 258)

- Figures 2, 6, and 8-11 (all are original figures / diagrams which the authors bear copyright ownership of and amended to be indicative of such).

And from requested additional content included in lines 808-830.
